# Association between Pulmonary Function and Body Composition in Children and Adolescents with and without Obesity

**DOI:** 10.3390/jcm11247410

**Published:** 2022-12-14

**Authors:** Mariana Simões Ferreira, Fernando Augusto Lima Marson, Vaneza Lira Waldow Wolf, Mariana Porto Zambon, Maria Ângela Reis de Góes Monteiro Antonio, José Dirceu Ribeiro, Roberto Teixeira Mendes

**Affiliations:** 1Department of Pediatrics, Faculty of Medical Sciences, University of Campinas, Campinas 13083-887, Brazil; 2Laboratory of Lung Function, Center of Investigation in Pediatrics, Faculty of Medical Sciences, University of Campinas, Campinas 13083-887, Brazil; 3Teaching and Research Center of Rede Mário Gatti, Campinas 13036-902, Brazil; 4Laboratory of Human and Medical Genetics, Post-Graduation Program in Health Sciences, São Francisco University, Bragança Paulista 12916-900, Brazil

**Keywords:** body composition, dysanapsis, lung function, obesity, spirometry

## Abstract

Lung function in children and adolescents with obesity must consider the coexistence of two complex and related phenomena: obesity and growth. The assessment of body composition can identify changes in respiratory dynamics arising, exclusively or jointly, from adiposity and lean body mass. This study aimed to compare pulmonary function and the dysanapsis indices of children and adolescents without asthma, with and without obesity, considering body composition, pubertal development, and physical activity practice. We performed a cross-sectional study with 69 participants, 41 (59.42%) of whom have obesity. All participants carried out spirometry and the assessment of, respectively, body composition by dual-energy X-ray absorptiometry, vital signs, pubertal development, and physical activity practice. In our data, the group with obesity had higher values of forced vital capacity (FVC) and lower values of the ratio between forced expiratory volume in one second and FVC (FEV_1_/FVC). Analyzing the entire sample, we found a positive correlation between FVC and a negative correlation between FEV_1_/FVC with fat mass markers. At the same time, inspiratory capacity, expiratory reserve volume, and peak expiratory flow were correlated with lean body mass markers. In addition, participants with obesity presented a lower dysanapsis index. In conclusion, children and adolescents with obesity showed increased FVC and reduced FEV_1_/FVC. Our findings are possibly related to the increase in fat mass, not to lean body mass. We hypothesize that these findings are associated with the dysanaptic growth pattern, which is higher in obesity, evidenced by the reduction of the dysanapsis index.

## 1. Introduction

Obesity is a chronic dysfunction that interferes with multiple systems of the human body, including the respiratory system and pulmonary function [1,2,3]. The literature, unlike for adults [4,5,6,7], has no consensus on how obesity modulates the pulmonary function of children and adolescents or about when excess weight starts to impair the respiratory system [8,9,10,11,12,13,14,15,16,17]. Thus, the critical analysis of this topic should consider the coexistence of two complex and related phenomena: obesity and growth.

Both obesity and growth act in the development of systems. For this reason, in the analysis of the pulmonary function of individuals with and without obesity, one must consider the transformations of the body during childhood and puberty [17]. Recent studies have returned to the concept of airway dysanapsis, which may portray the interference of obesity in respiratory system growth [18,19].

The interpretation of pulmonary function and growth in obesity is even more complex in the presence of diseases. The literature frequently associates the pulmonary function of children and adolescents with obesity and asthma since they present prevalence with parallel trajectories over the years and inflammatory and immunological aspects, and their causality relationship is not yet well defined [20,21,22,23].

Thus, this study aimed to compare the pulmonary function and dysanapsis indices (DIs) of children and adolescents without asthma, with and without obesity, considering their body composition, pubertal development, and physical activity practice.

## 2. Materials and Methods

### 2.1. Characterization of the Study Participants

Non-smoking participants without asthma, according to the International Study of Asthma and Allergies in Childhood (ISAAC) questionnaire [24], of both sexes, aged between four and 19 years old, were included. Initially, the sample consisted of 83 participants. Fourteen participants were excluded, and the final sample comprised 69 participants, 41 (59.42%) of whom had obesity.

Participants with obesity do follow-up in the outpatient clinic of obesity in children and adolescents of the Clinical Hospital of the University of Campinas (Unicamp) and receive multidisciplinary guidelines regarding the practice of physical activity and healthy eating. Participants of the control group volunteered to take part in the assessments, seeking the service after disclosure in social networks.

### 2.2. Evaluated Markers—Demographic, Clinical, and Laboratory

The participants and those responsible for them received prior guidance for participants to attend the study location with light clothes, without metals (such as buttons or zippers), and having only a light meal six hours before the evaluations. Participants were told not to perform strenuous physical activity in the 24 h prior to the assessments. In the study, sex and ethnicity (race) were obtained by interview.

#### 2.2.1. Anthropometric Markers and Vital Signs

On the assessment day, participants remained resting for at least ten minutes. Subsequently, the anthropometric measurements of body weight and standing height were carried out, and the vital signs of heart rate (HR), respiratory rate (RR), peripheral oxygen saturation (SpO_2_), and systolic (SBP) and diastolic (DBP) blood pressure were measured. The anthropometric data were used to calculate the body mass index (BMI)—weight/height^2^—and its distribution in the percentiles and z-score, following the criteria of the World Health Organization (WHO) for individuals up to 19 years [25]. The data were estimated in WHO AnthroPlus software (https://www.who.int/growthref/tools/en/; Geneva, Switzerland—accessed on 6 November 2022). The normal value was a BMI z-score ranging from −2 to +1. Individuals with obesity were considered those with a BMI z-score > +2. Mean blood pressure (MBP) was calculated from SBP and DBP by the formula: MBP = [SBP + (DBP × 2)]/3 [26].

#### 2.2.2. Pulmonary Function—Spirometry and Dysanapsis Index

Pulmonary function was assessed in the spirometer MasterScreen^TM^ Pneumo (Jaeger; Wüzburg, Germany), integrated with JLab version 5.20 (Erich Jaeger, Inc., Wüzburg, Germany), following the specifications of the European Respiratory Society (ERS) and American Thoracic Society (ATS) [27]. The participants underwent spirometry standing, keeping their heads upright, and using a nose clip. The following markers were analyzed: (i) forced vital capacity (FVC); (ii) forced expiratory volume in one second (FEV_1_); (iii) FEV_1_/FVC; (iv) expiratory reserve volume (ERV); (v) average expiratory flow (FEF25–75%); (vi) peak expiratory flow (PEF); and (vii) inspiratory capacity (IC). FVC, FEV_1_, FEV_1_/FVC, and FEF25–75% were analyzed by z-score and percentile, using the references from Quanjer et al. (2012) from the software of the Global Lung Initiative. 

The absolute values of pulmonary function markers were used in the calculation of the DI, proposed by Mead (DI1) (1980), to quantify the ratio between lung and airway size. DI was calculated by the ratio between forced expiratory flow at 50% (FEF50%) of the FVC, FVC, and elastic recoil pressure at 50% of the FVC (Pst 50%) by the formula: DI1 = FEF50%/(FVC × Pst 50%) [28]. The Pst 50% was obtained by the formula: 6.3038 − (0.056 × age). In this study, the calculation of the DI, proposed by Tager et al. (DI2) (1986), was also measured by the formula: DI2 = FEF25–75%/FVC [29].

#### 2.2.3. Dual-Energy X-ray Absorptiometry

Body composition was measured by dual-energy X-ray absorptiometry (DXA) in the equipment iDXA (GE Healthcare Lunar, Madison, WI, USA) with fan beam detectors. The data were processed and analyzed in enCoreTM (2011), version 13.6 (GE Healthcare Lunar), with the inclusion of the absolute value and normality adjustment for age, sex, and ethnicity (race). Some of the participants with obesity exceeded the area of assessment of DXA; thus, the whole-body technique with mirroring of the left arm was used to standardize the measures without a bias between the techniques. The DXA markers assessed were: (i) fat mass and lean body mass (trunk, android, gynoid, and total); (ii) total body mass; (iii) total fat percentage; and (iv) fat-free mass.

#### 2.2.4. Physical Activity and Pubertal Development

This study collected data considering the practice of scheduled and unscheduled physical activities by applying the international physical activity questionnaire (IPAQ) [30]. According to the responses, participants were classified as: very active, active, irregularly active, or sedentary. Later, for homogenization of the groups, the classifications were divided into very active and active or irregularly active and sedentary. Additionally, pubertal development was also assessed by Marshall and Tanner’s criteria of pubic hair and genitalia for boys and pubic hair and breasts for girls [31,32].

### 2.3. Statistical Analysis

The descriptive analysis is presented by relative and absolute frequencies for categorical data and mean ± standard deviation and median (25th percentile and 75th percentile) for the data with the numerical distribution. The following techniques evaluated the normality of the numerical data: (i) analysis of descriptive measures for central trend; (ii) graphic method (normal Q-Q plot, Q-Q plot without trend, and boxplot); and (iii) method by statistical testing (normality tests): Kolmogorov–Smirnov and Shapiro–Wilk tests.

The association between the groups with and without obesity—the control group (independent variable) and the numeric markers (dependent variable) for the two groups—was examined by the Mann–Whitney U test for independent samples or by a *t*-test for independent samples, depending on the data distribution. The same statistical tests were applied to compare the distribution of the numerical data according to sex in each group (obesity versus control group). Pearson correlation and Spearman correlation tests were used in the correlation analysis, depending on the data distribution. In the comparison between categorical variables, Fisher’s exact and chi-square tests were applied. 

The statistical analysis was performed in the Statistical Package for the Social Sciences (IBM Corp. Released 2017. IBM SPSS Statistics for Windows, Version 25.0. IBM Corp., Armonk, NY, USA). The analyses considered the alpha value of 0.05. No technique was used to handle the adjustments for “missing data”. All the data were collected from all participants for the measures of pulmonary function and body composition.

## 3. Results

Table 1 presents the distribution of groups for the anthropometric data. This study showed no differences between the groups with and without obesity (control group) for height, age, and sex. As expected, weight, BMI, and BMI z-score (BMI-z) were higher in the group with obesity (*p* < 0.05). In addition, ethnic (race) differences between the groups were found (*p* = 0.038). There were no significant differences in DI2 (*p* = 0.065) between groups; however, participants from the obesity group presented lower DI1 (*p* = 0.031). 

Table 2 presents the distribution of participants regarding physical activity practice and shows no difference between the groups according to obesity. However, differences between the groups for some markers were found for clinical signs. The group with obesity showed the highest values of RR, SBP, DBP, and MBP compared to the group without obesity (control group) (Table 3). HR and SpO_2_ were equal between the groups. 

Figure 1 shows data concerning pubertal development according to genitalia and breasts (*p* = 0.138)/pubic hair (*p* = 0.089), and no difference was found between the groups with and without obesity.

Considering the entire sample, we found a strong/moderate positive correlation between BMI-z and fat mass markers (Figure 2). However, a significative correlation was not observed for lean body mass markers (*p* > 0.05). Trunk fat mass and BMI-z had a CC = 0.791 (*p* < 0.001), while trunk lean mass and BMI-z had a CC = 0.153 (*p* = 0.214). Android fat mass and BMI-z had a CC = 0.816 (*p* < 0.01), while android lean mass and BMI-z had a CC = 0.196 (*p* = 0.109). Gynoid fat mass and BMI-z had a CC = 0.750 (*p* < 0.01), while gynoid lean mass and BMI-z had a CC = 0.163 (*p* = 0.184). Finally, total fat mass and BMI-z had a CC = 0.775 (*p* < 0.001), while total lean mass and BMI-z had a CC = 0.154 (*p* = 0.209). When we analyzed correlation data considering the division by groups with and without obesity (control group), we found only a weak correlation between trunk fat mass and BMI-z (CC = 0.310; *p* = 0.048) and between android fat mass and BMI-z (CC = 0.394; *p* = 0.011), both in the group of participants with obesity.

Table 4 presents the correlation between pulmonary function markers and body composition variables. When we considered the entire sample, we observed significant correlations between fat mass markers and FVC z-score (trunk (CC = 0.329; *p* = 0.006), android (CC = 0.338; *p* = 0.005), gynoid (CC = 0.301; *p* = 0.012), total (CC = 0.315; *p* = 0.008), and fat percentage (CC = 0.359; *p* = 0.002)), FVC percentile (trunk (CC = 0.330; *p* = 0.006), android (CC = 0.339; *p* = 0.004), gynoid (CC = 0.302; *p* = 0.012), total (CC = 0.317; *p* = 0.008), and fat percentage (CC = 0.358; *p* = 0.002)), and FEV_1_/FVC z-score (trunk (CC = −0.256; *p* = 0.034), android (CC = −0.260; *p* = 0.031), total (CC = −0.244; *p* = 0.043), and fat percentage (CC = −0.263; *p* = 0.029)). 

Additionally, we found significant correlations between lean mass markers and IC (trunk (CC = 0.341; *p* = 0.006), android (CC = 0.338; *p* = 0.007), gynoid (CC = 0.334; *p* = 0.007), total (CC = 0.322; *p* = 0.010), and fat-free mass (CC = 0.340; *p* = 0.006)), ERV (trunk (CC = −0.262; *p* = 0.036), android (CC = −0.253; *p* = 0.044), gynoid (CC = −0.271; *p* = 0.030), and fat-free mass (CC = −0.266; *p* = 0.034)), and PEF (trunk (CC = 0.287; *p* = 0.018), android (CC = 0.258; *p* = 0.033), gynoid (CC = 0.268; *p* = 0.027), total (CC = 0.291; *p* = 0.016), and fat-free mass (CC = 0.276; *p* = 0.023)) (Table 4).

When analyzing the correlations considering groups with and without obesity (control group), we found different results. In the group with obesity, the significant correlations with fat mass markers were with IC (trunk (CC = 0.352; *p* = 0.033)) and PEF (trunk (0.472; *p* = 0.002), android (CC = 0.457; *p* = 0.003), gynoid (CC = 0.458; *p* = 0.003), and total (CC = 0.467; *p* = 0.002)). At the same time, we found significant correlations between lean mass and IC (trunk (CC = 0.401; *p* = 0.014), android (CC = 0.390; *p* = 0.017), gynoid (CC = 0.369; *p* = 0.025), total (CC = 0.383; *p* = 0.019), and fat-free mass (CC = 0.411; *p* = 0.011)), and PEF (trunk (CC = 0.519; *p* = 0.001), android (CC = 0.497; *p* = 0.001), gynoid (CC = 0.529; *p* < 0.001), total (CC = 0.510; *p* = 0.001), and fat-free mass (CC = 0.478; *p* = 0.002)) (Table 4). In the group without obesity (control group), we found a significant correlation between trunk fat mass and PEF (trunk (CC = −0.389; *p* = 0.045)), and we also found significant correlations between lean mass markers and FVC z-score (gynoid (CC = −0.406; *p* = 0.032), total (−0.381; *p* = 0.046), and fat-free mass (−0.424; *p* = 0.024)), FVC percentile (gynoid (CC = −0.414; *p* = 0.029), total (CC = −0.386; *p* = 0.042), and fat-free mass (CC = −0.431; *p* = 0.022)), and ERV (gynoid (CC = −0.458; *p* = 0.018)) (Table 4).

In the evaluation of pulmonary function, participants with obesity presented higher FVC (z-score (*p* = 0.002) and percentile (*p* = 0.03)) and lowered FEV_1_/FVC (z-score (*p* = 0.02) and percentile (*p* = 0.049)) (Table 5). 

Table 6 presents the differences in fat mass markers between groups with and without obesity, considering a subdivision of groups by sex. All variables were significantly higher in the group with obesity for both males and females (*p* < 0.001).

Nevertheless, when considering the differences in lean mass markers between groups, the differences remain significant only in the female group, as is shown in Table 7. Besides the fat markers, girls with obesity also had significantly higher values in the mean of lean mass markers (trunk: *p* = 0.05; android: *p* = 0.033; gynoid: *p* = 0.044; total: *p* = 0.028; and fat-free mass: *p* = 0.023).

## 4. Discussion

In the analysis of anthropometric data, it is interesting to note that BMI-z was an accurate variable to assess the adiposity of participants. The literature points to numerous limitations regarding the use of BMI since it quantifies mass and not fat and may overestimate and classify as overweight or obese individuals with high lean mass [33,34]. However, in the population of children and adolescents evaluated in our study, BMI-z presented a strong correlation with the markers of fat mass (trunk, android, gynoid, and fat percentage) and no correlation with lean body mass markers. Therefore, the classification of the groups with and without obesity was faithful to body composition, which confirms the difference between the groups for fat mass and not for lean body mass.

Although the relationship between obesity and physical inactivity is clear, no difference was found between participants with and without obesity for physical activity practice. We believe this finding is related to three possibilities: (i) a small sample size with a high probability of type II error (false negative result); (ii) a group with obesity receiving multidisciplinary outpatient follow-up with physical educators who emphasize the importance of physical activities—individuals with obesity are in the process of transition of life habits, and possibly for this reason there was no difference in the practice of physical activity among children and adolescents with and without obesity; and (iii) high prevalence of physical inactivity or irregular activity among children and adolescents, whether obese (56.1%) or not (42.9%), a fact already discussed in the literature [35,36,37,38]. The prevalence of physical inactivity was high, noting that the IPAQ assessment addresses scheduled and unscheduled activities and time sitting. The widespread use of screens [39,40], the short time of physical activity at school, the lack of adequate and safe spaces and equipment (e.g., sports courts and swimming pools) in Brazil, and inadequate habits from parents, which culturally follow this inactive model, relegate children to their house and the sedentary lifestyle, without a satisfactory energy expenditure [41,42].

The increase in SBP, DBP, and MBP between the groups, even though these are statistically equal regarding height and age, shows a possible overload of the cardiovascular system in obesity. Excess adipose tissue increases metabolic demands; for this reason, the body requires more blood supply, with a concomitant increase in cardiac activity [43]. The described excessive demand increases the risk of cardiovascular comorbidity and death [44]. 

Obesity also presented a greater RR value. The increased metabolic demand, mentioned earlier, implies a greater need for oxygen, with the consequent need for greater lung volume. However, the increased fat deposition, especially around the thorax and abdomen, generates a mechanical barrier that hinders thoracic expandability, and this could explain why an individual with obesity requires more respiratory incursions per minute [20,45].

This study conducted a comparative analysis of pulmonary function and body composition by DXA, which allowed us to verify more accurately the respiratory variables related to obesity. When comparing groups, we found that individuals with obesity had significantly higher values in FVC and, consequently, significantly lower values in FEV_1_/FVC. Analyzing the entire sample, we have found that FVC was positively correlated with fat mass (total, trunk, android, gynoid, and fat percentage) and that FEV_1_/FVC was negatively correlated with the mentioned markers, except gynoid fat mass. Other variables correlated with muscular strength and were assessed by lean body mass markers (trunk, android, gynoid, and total) with IC and PEF. 

However, when we analyzed the correlations grouping the sample with and without obesity, these differences did not hold. We believe that the sample size, after the group division, could interfere with the power of statistical analysis.

Another fact that stood out was the positive correlation between FVC and fat mass markers; in the comparison between the groups, this variable was higher in participants with obesity. This group also presented the lowest FEV_1_/FVC and a negative correlation with fat mass measures. According to the literature, this finding was present in studies with children and adolescents and was not found in adults [5,8,9,13,14,15,22,46]. Thus, this fact guides us to associate it with the growth period. 

Within this perspective, some previous studies justified findings similar to dysanapsis of the airways, which is the disproportionate growth between the pulmonary parenchyma and the airways, with an increase in lung volume different from the increase in the caliber of the airways [22]. This growth pattern is physiologically influenced by sex [47]; however, the relationship of obesity with dysanaptic growth has been studied [18,19,48]. This study found differences between the DIs, assessed by one method [28,29]. Besides that, in our view, the lower difference is justified by the careful selection/exclusion of participants with respiratory symptoms. Even without a great change in DI, the increased FVC, reduced FEV_1_/FVC, and FEV_1_ with no difference in participants with obesity directs the findings to the dysanaptic pattern in this group of individuals. 

One of the differences of our study was the exclusion of participants with asthma, differing from previous studies associating asthma and obesity, even if independent. In addition, the body composition analysis has allowed us to confirm that the increase in FVC and reduction in FEV_1_/FVC is related to adiposity and not to an increase in strength of participants with obesity, because of early maturation, as was discussed in the literature [46]. This analysis has shown that the stress-dependent variables—thus positively influenced by increasing lean body mass—are PEF, IC, and ERV. 

The body composition analysis was a differential to understand the pulmonary function of children and adolescents with and without obesity, basing its clinical applicability in the pediatric area, especially in the care of individuals with obesity, since obesity and growth are complex phenomena that can occur concurrently. It is important to note that differences in body composition can result in lung function implications. As we observed, both boys and girls with obesity had significantly higher fat markers, as expected. However, girls with obesity also had significantly higher lean mass markers when compared to girls without obesity. Studies show that obesity can contribute to the early onset of puberty in girls [49,50,51,52]. The earlier pubertal development may interfere with the growth and development of muscular mass, which can explain higher values of lean mass in girls with obesity.

To the best of our knowledge, this is the only study that established a correlation between lean body mass and fat mass variables with the pulmonary function of children and adolescents without asthma.

The present study had some limitations. The number of participants may have been insufficient to identify the differences between both DIs. In addition, we did not analyze the correlations between groups with and without obesity considering sex differences because the subgroups would be too small, compromising the relevance of the statistical analysis. 

## 5. Conclusions

We conclude that the relationship between obesity and pulmonary function in children and adolescents is complex and that it is essential to reduce the confounding biases, to understand which changes are obesity-related and which are growth-related. Participants with obesity showed increased FVC and reduced FEV_1_/FVC. The findings are related to the increase in fat mass, with no relation to lean body mass. We hypothesize that these findings are associated with the dysanaptic growth pattern, which is higher in obesity, evidenced by the reduction of dysanapsis index.

## Figures and Tables

**Figure 1 jcm-11-07410-f001:**
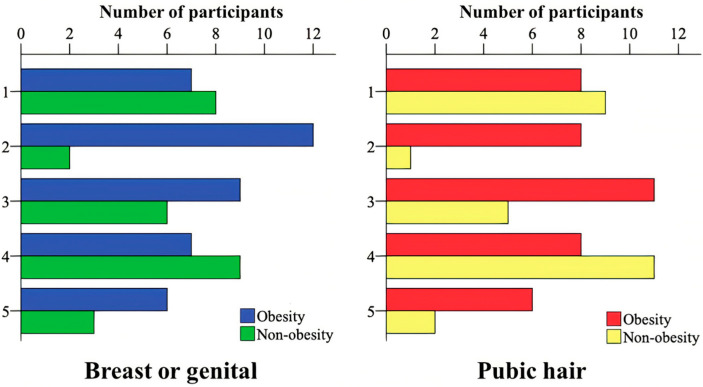
Distribution of pubertal development according to genitalia and breasts (*p* = 0.138) and pubic hair (*p* = 0.089) between the groups with and without obesity.

**Figure 2 jcm-11-07410-f002:**
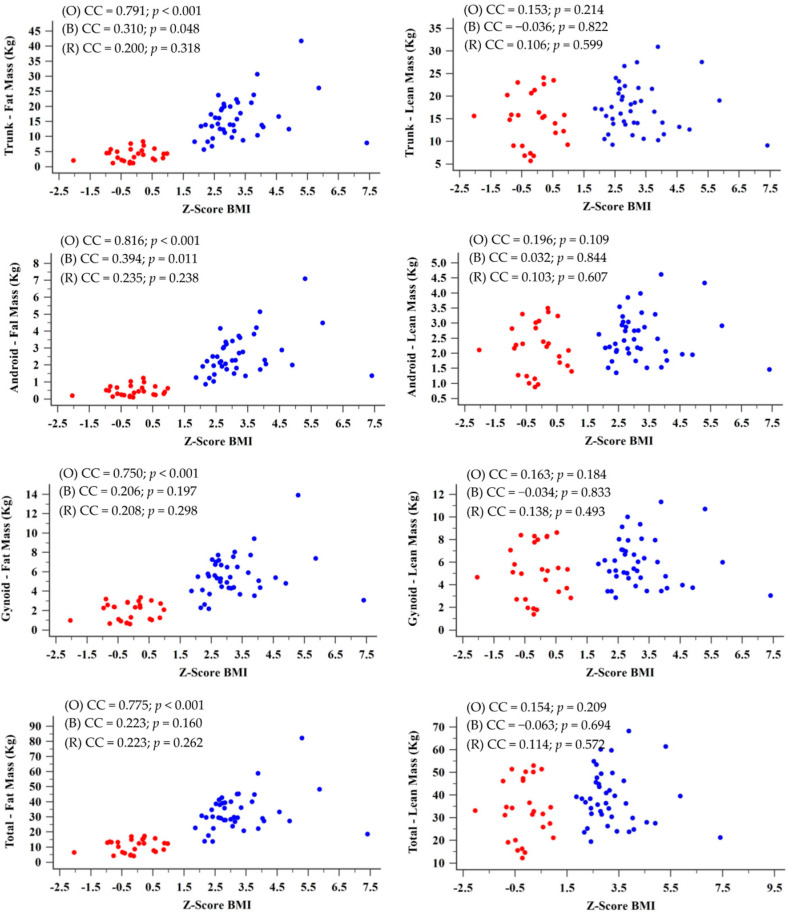
Correlation between the body mass index (BMI) z-score and the variables of fat mass and lean body mass. Blue (B) indicates participants with obesity, and red (R) indicates participants without obesity (control group). CC, correlation coefficient. The statistical analyses were carried out based on Spearman’s rank correlation coefficient. Alpha = 0.05.

**Table 1 jcm-11-07410-t001:** Distribution and comparison of sex, ethnicity (race), anthropometric markers, and dysanapsis indexes between groups with and without obesity.

Marker	Group	Obesity (*n* = 41)	Control (*n* = 28)	*p*
Sex ^1^	Male	14 (34.1%)	14 (50%)	0.188
	Female	27 (65.9%)	14 (50%)	
Ethnicity (race) ^2^	White	25 (61%)	25 (85.7%)	**0.038**
	Black	8 (19.5%)	1 (3.6%)	
	Mixed race *	8 (19.5%)	2 (7.1%)	
	Asian	−	1 (3.6%)	
Height ^3^ (cm)		153.27 ± 13.86; 155.50 (142.70 to 163.25)	152.65 ± 22.75; 161.90 (134.50 to 171.90)	0.460
Weight ^4^ (kg)		74.04 ± 23.67; 69.10 (58.35 to 89.45)	44.98 ± 17.22; 49.30 (29.27 to 61.53)	**<0.001**
BMI ^3^ (kg/m^2^)		30.85 ± 6.63; 29.92 (27.24 to 33.44)	18.30 ± 2.54; 18.84 (15.57 to 20.3)	**<0.001**
BMI Z-score ^3^		3.25 ± 1.09; 3 (2.57 to 3.73)	−0.16 ± 0.68; −0.20 (−0.63 to 0.22)	**<0.001**
Age ^3^ (years)		11.98 ± 3.62; 11.74 (8.99 to 14.94)	12.97 ± 4.91; 14.18 (8.72 to 18.05)	0.213
DI1 ^4^		0.20 ± 0.40; 0.19 (0.16 to 0.23)	0.22 ± 0.50; 0.19 to 0.26)	**0.031**
DI2 ^4^		0.98 ± 0.24; 0.95 (0.81 to 1.14)	1.09 ± 0.27; 1.13 (0.87 to 1.25)	0.065

*n*, number of participants; cm, centimeters; BMI, body mass index; kg, kilograms; kg/m^2^, kilograms/square meter; DI1, dysanapsis index 1; DI2, dysanapsis index 2. Categorical data are presented in absolute frequency (relative frequency). Numerical data are presented by mean ± standard deviation; median (25th percentile to 75th percentile). In the statistical analysis of the data, the following tests were used: ^1^, chi-square test; ^2^, Fisher’s exact test; ^3^, Mann–Whitney test for independent samples; ^4^, *t*-test for independent samples. *, *Pardos* (multiracial background). Alpha = 0.05. Data with significant *p* (≤0.05) are presented in bold.

**Table 2 jcm-11-07410-t002:** Classification and comparison of the groups regarding the practice of physical activities, according to the IPAQ, considering the complete (A) and synthesized (B) classifications.

A ^1^	Obesity	Control	*p*
Sedentary	1 (2.4%)	−	0.777
Irregularly active A	8 (19.5%)	3 (10.7%)	
Irregularly active B	14 (34.1%)	9 (32.1%)	
Active	6 (14.6%)	6 (21.4%)	
Very active	12 (29.3%)	10 (35.7%)	
**B ^2^**	**Obesity**	**Control**	** *p* **
Sedentary or irregularly active	23 (56.1%)	12 (42.90%)	0.280
Active or very active	18 (43.9%)	16 (57.1%)	

In the statistical analysis of the data, the following tests were used: ^1^, Fisher’s exact test; ^2^, chi-square test. IPAQ, international physical activity questionnaire. Alpha = 0.05. Categorical data are presented in absolute frequency (relative frequency).

**Table 3 jcm-11-07410-t003:** Comparison between the vital signs of participants with and without obesity.

Variable	Obesity	Control	*p*
Heart rate ^1^	85.83 ± 15.65; 87 (73 to 97.50)	81.11 ± 13.70; 82 (70.25 to 90.50)	0.200
Respiratory rate ^1^	21.44 ± 3.97; 22 (18 to 23.50)	18.43 ± 4.51; 18 (15.25 to 22.75)	**0.005**
SpO_2_ ^2^	97.44 ± 1.21; 98 (97 to 98)	97.50 ± 0.92; 97 (97 to 98)	0.798
**Blood pressure**	**Obesity**	**Control**	** *p* **
Systolic ^1^	118.39 ± 14.03; 120 (87 to 147)	102.50 ± 12.23; 103 (86 to 124)	**<0.001**
Diastolic ^2^	78.12 ± 10.54; 78 (71 to 80)	67.89 ± 7.36; 70 (60 to 70)	**<0.001**
Mean ^2^	91.54 ± 10.96; 92.33 (85 to 96.50)	79.43 ± 8.28; 80.17 (70.17 to 86)	**<0.001**

SpO_2_, peripheral oxygen saturation. Numerical data are presented by mean ± standard deviation; median (25th percentile to 75th percentile). In the statistical analysis of the data, the following tests were used: ^1^, *t*-test for independent samples; ^2^, Mann–Whitney test. Alpha = 0.05. Data with significant *p* (≤0.05) are presented in bold.

**Table 4 jcm-11-07410-t004:** Correlation between pulmonary function markers and body composition, respectively, evaluated by spirometry and dual-energy X-ray absorptiometry.

Markers		Fat Mass					Lean Body Mass				
Overall		Trunk	Android	Gynoid	Total	Fat Percentage	Trunk	Android	Gynoid	Total	Fat-Free Mass
FVC z-score	CC	**0.329**	**0.338**	**0.301**	**0.315**	**0.359**	−0.033	0.011	−0.074	−0.042	−0.073
	*p*	**0.006**	**0.005**	**0.012**	**0.008**	**0.002**	0.791	0.926	0.548	0.731	0.551
FVC—percentile	CC	**0.330**	**0.339**	**0.302**	**0.317**	**0.358**	−0.031	0.014	−0.072	−0.040	−0.071
	*p*	**0.006**	**0.004**	**0.012**	**0.008**	**0.002**	0.802	0.911	0.556	0.744	0.561
FEV_1_/FVC z-score	CC	**−0.256**	**−0.260**	−0.236	**−0.244**	**−0.263**	0.029	−0.010	0.022	0.025	0.030
	*p*	**0.034**	**0.031**	0.051	**0.043**	**0.029**	0.811	0.937	0.855	0.840	0.805
Inspiratory capacity	CC	0.162	0.133	0.152	0.144	−0.063	**0.341**	**0.338**	**0.334**	**0.322**	**0.340**
	*p*	0.206	0.298	0.235	0.259	0.624	**0.006**	**0.007**	**0.007**	**0.010**	**0.006**
ERV	CC	−0.090	−0.063	−0.066	−0.064	0.091	**−0.262**	**−0.253**	**−0.271**	−0.241	**−0.266**
	*p*	0.481	0.621	0.605	0.616	0.477	**0.036**	**0.044**	**0.030**	0.055	**0.034**
PEF	CC	0.065	0.067	0.081	0.072	−0.092	**0.287**	**0.258**	**0.268**	**0.291**	**0.276**
	*p*	0.599	0.587	0.509	0.559	0.457	**0.018**	**0.033**	**0.027**	**0.016**	**0.023**
**Obesity**											
FVC z-score	CC	0.210	0.209	0.233	0.184	−0.019	0.183	0.194	0.166	0.160	0.142
	*p*	0.188	0.189	0.143	0.249	0.907	0.253	0.225	0.300	0.300	0.377
FVC—percentile	CC	0.216	0.215	0.239	0.190	−0.019	0.187	0.198	0.170	0.164	0.145
	*p*	0.176	0.177	0.133	0.234	0.906	0.242	0.214	0.289	0.306	0.365
FEV_1_/FVC z-score	CC	0.023	0.002	0.049	0.061	0.065	0.008	−0.032	0.013	0.005	0.022
	*p*	0.888	0.988	0.760	0.703	0.686	0.961	0.842	0.937	0.977	0.890
Inspiratory capacity	CC	**0.352**	0.295	0.265	0.290	−0.213	**0.401**	**0.390**	**0.369**	**0.383**	**0.411**
	*p*	**0.033**	0.076	0.114	0.081	0.205	**0.014**	**0.017**	**0.025**	**0.019**	**0.011**
ERV	CC	−0.175	−0.126	−0.077	−0.115	0.175	−0.171	−0.165	−0.133	−0.149	−0.184
	*p*	0.294	0.450	0.645	0.491	0.292	0.303	0.323	0.427	0.372	0.270
PEF	CC	**0.472**	**0.457**	**0.458**	**0.467**	0.017	**0.519**	**0.497**	**0.529**	**0.510**	**0.478**
	*p*	**0.002**	**0.003**	**0.003**	**0.002**	0.914	**0.001**	**0.001**	**<0.001**	**0.001**	**0.002**
**Without obesity**											
FVC z-score	CC	−0.184	−0.167	−0.208	−0.208	0.285	−0.355	−0.287	**−0.406**	**−0.381**	**−0.424**
	*p*	0.349	0.396	0.288	0.289	0.141	0.064	0.138	**0.032**	**0.046**	**0.024**
*Continue*											
FVC—percentile	CC	−0.187	−0.169	−0.209	−0.208	0.286	−0.361	−0.292	**−0.414**	**−0.386**	**−0.431**
	*p*	0.342	0.391	0.286	0.287	−0.140	0.059	0.131	**0.029**	**0.042**	**0.022**
FEV_1_/FVC z-score	CC	−0.113	−0.111	−0.061	−0.079	−0.251	0.171	0.136	0.130	0.174	0.168
	*p*	0.568	**0.573**	0.757	0.690	0.198	0.385	0.489	0.509	0.375	0.394
Inspiratory capacity	CC	0.095	**0.050**	0.191	0.127	−0.092	0.272	0.294	0.364	0.316	0.309
	*p*	0.645	0.810	0.350	0.536	0.655	0.179	0.144	0.067	0.116	0.124
ERV	CC	−0.257	−0.218	−0.274	−0.247	0.028	−0.355	−0.359	**−0.458**	−0.379	−0.377
	*p*	0.205	0.284	0.176	0.223	0.893	0.075	0.072	**0.018**	0.056	0.058
PEF	CC	**−0.389**	−0.351	−0.279	−0.320	−0.367	0.003	−0.049	−0.003	0.032	0.054
	*p*	**0.045**	0.073	0.158	0.103	0.060	0.988	0.806	0.988	0.873	0.789

CC, correlation coefficient; FVC, forced vital capacity; FEV_1_/FVC, ratio between forced expiratory volume in one second and forced vital capacity; ERV, expiratory reserve volume; PEF, peak expiratory flow. Spearman’s rank correlation coefficient was used in the statistical analysis of the data. Alpha = 0.05. Data with significant *p* (≤0.05) are presented in bold.

**Table 5 jcm-11-07410-t005:** Differences between pulmonary function markers in groups of participants with and without obesity.

Markers	Obesity	Control	*p*
FVC z-score	0.43 ± 0.62; 0.41 (0.025 to 0.82)	−0.20 ± 0.92; −0.23 (−0.98 to 0.49)	**0.002** ^1^
FVC—percentile	64.32 ± 19.77; 66 (51 to 79)	43.46 ± 28.89; 41 (16.50 to 68.50)	**0.003** ^2^
FEV_1_ z-score	0.39 ± 0.69; 0.36 (0.01 to 0.79)	0.06 ± 0.96; 0.05 (−0.66 to 0.77)	0.125 ^1^
FEV_1_—percentile	63.24 ± 21.54; 64 (50.50 to 78.50)	51.86 ± 29.16; 51.50 (25.75 to 77.75)	0.124 ^2^
FEV_1_/FVC z-score	−0.10 ± 1.01; −0.33 (−0.76 to 0.49)	0.54 ± 1.14; 0.75 (−0.50 to 1.45)	**0.020** ^1^
FEV_1_/FVC percentile	45.90 ± 29.79; 37 (22.50 to 69)	61.37 ± 33.48; 71 (27.25 to 92.75)	**0.049** ^2^
FEF25–75% z-score	−0.13 ± 0.97; −0.15 (−0.73 to 0.63)	−0.09 ± 1.08; −0.02 (−0.97 to 0.64)	0.885 ^1^
FEF25–75% percentile	45.95 ± 28.43; 44 (23.50 to 73.50)	47.64 ± 31.14; 49.50 (16.75 to 73.75)	0.869 ^2^
Inspiratory capacity	103.34 ± 38.77; 97.70 (71.75 to 140.45)	101.65 ± 31.79; 100.25 (80.73 to 124.10)	0.791 ^2^
Expiratory reserve volume	117.44 ± 84.25; 103.65 (49.13 to 167.73)	123.87 ± 113.40; 111.45 (53.48 to 149.03)	0.796 ^1^
Peak expiratory flow	87.64 ± 15.72; 88 (76.05 to 96.85)	89.28 ± 14.36; 90.40 (79.80 to 100.90)	0.665 ^1^

FVC, forced vital capacity; FEV_1_/FVC, ratio between forced expiratory volume in one second and forced vital capacity. Numerical data are presented by mean ± standard deviation; median (25th percentile to 75th percentile). In the statistical analysis of the data, the following tests were used: ^1^, *t*-test for independent samples; ^2^, Mann–Whitney test for independent samples. Alpha = 0.05. Data with significant *p* (≤0.05) are presented in bold.

**Table 6 jcm-11-07410-t006:** Association between body composition (fat mass) evaluated by dual-energy X-ray absorptiometry in groups of participants with and without obesity and according to sex.

Trunk Fat Mass	Obesity	Control	*p*
Male	16.96 ± 5.80; 18.30 (11.20 to 21.56)	3.79 ± 2.22; 3.00 (2.17 to 5.19)	**<0.001**
Female	15.47 ± 7.56; 13.80 (11.26 to 17.77)	4.17 ± 2.19; 4.32 (1.77 to 5.85)	**<0.001**
*p*	0.204	0.635	
**Android Fat Mass**	**Obesity**	**Control**	** *p* **
Male	2.84 ± 1.06; 3.07 (1.73 to 3.75)	0.47 ± 0.34; 0.34 (0.23 to 0.70)	**<0.001**
Female	2.48 ± 1.32; 2.15 (1.75 to 2.78)	0.49 ± 0.29; 0.47 (0.20 to 0.74)	**<0.001**
*p*	0.176	0.874	
**Gynoid Fat Mass**	**Obesity**	**Control**	** *p* **
Male	5.76 ± 1.63; 5.66 (4.22 to 7.34)	1.68 ± 0.80; 1.28 (1.02 to 2.42)	**<0.001**
Female	5.55 ± 2.38; 5.36 (4.34 to 6.51)	2.26 ± 1.11; 2.47 (0.86 to 3.09)	**<0.001**
*p*	0.488	0.246	
**Total Fat Mass**	**Obesity**	**Control**	** *p* **
Male	34.50 ± 9.19; 37.23 (25.82 to 42.26)	10.09 ± 4.26; 8.56 (6.57 to 13.75)	**<0.001**
Female	32.63 ± 13.95; 29.60 (27.24 to 38.02)	11.56 ± 5.06; 12.60 (5.65 to 15.28)	**<0.001**
*p*	0.320	0.667	
**Fat Percentage**	**Obesity**	**Control**	** *p* **
Male	43.64 ± 5.74; 43.07 (39.51 to 46.61)	20.87 ± 4.45; 21.20 (17.61 to 23.86)	**<0.001**
Female	46.03 ± 4.21; 46.12 (44.62 to 48.17)	28.00 ± 5.33; 27.10 (24.86 to 32.39)	**<0.001**
*p*	0.108	**0.001**	

Numerical data are presented by mean ± standard deviation; median (25th percentile to 75th percentile). The Mann–Whitney test for independent samples was applied in the statistical analysis of the data. Alpha = 0.05. Data with significant *p* (≤0.05) are presented in bold.

**Table 7 jcm-11-07410-t007:** Association between body composition (lean body mass) evaluated by dual-energy X-ray absorptiometry in groups of participants with and without obesity and according to sex.

Trunk Lean Body Mass	Obesity	Control	*p*
Male	19.47 ± 5.34; 20.10 (15.90 to 23.49)	17.01 ± 6.53; 20.48 (11.15 to 22.78)	0.286
Female	16.15 ± 5.16; 14.99 (11.59 to 19.04)	12.89 ± 4.27; 15.02 (8.50 to 15.87)	**0.050**
*p*	0.060	0.061	
**Android Lean Body Mass**	**Obesity**	**Control**	** *p* **
Male	2.87 ± 0.73; 2.90 (2.37 to 3.40)	2.41 ± 0.93; 2.83 (1.51 to 3.26)	0.165
*Continue*			
Female	2.37 ± 0.78; 2.18 (1.76 to 2.87)	1.84 ± 0.61; 2.13 (1.21 to 2.32)	**0.033**
*p*	0.054	0.069	
**Gynoid Lean Body Mass**	**Obesity**	**Control**	** *p* **
Male	6.83 ± 2.12; 6.99 (5.27 to 8.34)	5.90 ± 2.76; 7.42 (3.22 to 8.31)	0.323
Female	5.52 ± 2.05; 5.20 (3.75 to 6.67)	4.21 ± 1.64; 5.05 (2.52 to 5.41)	**0.044**
*p*	0.061	0.062	
**Total Lean Body Mass**	**Obesity**	**Control**	** *p* **
Male	43.51 ± 11.95; 44.20 (35.18 to 53.85)	37.98 ± 14.81; 46.32 (24.50 to 50.61)	0.287
Female	35.89 ± 11.35; 34.18 (26.33 to 40.92)	27.89 ± 9.04; 31.57 (18.24 to 34.66)	**0.028**
*p*	0.052	0.041	
**Fat-Free Mass**	**Obesity**	**Control**	** *p* **
Male	44.86 ± 12.59; 45.19 (35.61 to 55.29)	39.52 ± 15.54; 48.87 (25.67 to 51.34)	0.327
Female	37.19 ± 11.70;35.95 (27.56 to 41.48)	28.59 ± 9.58; 32.15 (18.24 to 36.67)	**0.023**
*p*	0.060	**0.036**	

Numerical data are presented by mean ± standard deviation; median (25th percentile to 75th percentile). The *t*-test for independent samples was applied in the statistical analysis of the data. Alpha = 0.05. Data with significant *p* (≤0.05) are presented in bold.

## Data Availability

The crude data presented in this study will be available on request.

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
