# Peer review of "Association between Pulmonary Function and Body Composition in Children and Adolescents with and without Obesity"

_jcm, 2022, doi:10.3390/jcm11247410_

Round 1

Reviewer 1 Report

In the manuscript entitled “Association between pulmonary function and body composition in children and adolescents with and without obesity”, the authors presented data from control and obese participants. The study should be strengthened with new and more relevant analyses. Additionnal analyses are needed.

Material and methods

The authors should check the number of participants that varied from 69 (line 55) to 70. In table 1, the authors mentioned 41 participants with obesity and 29 controls whereas in table 2 only 28 controls are studied.

Line 53: The authors mentioned that the study was performed in both sex. In Table 1, data from female should be added.

Line 76, z score should be defined and explain why they use it. The authors should explain the interest of using z-score and why it appears to be negative in the control group.

In general, comparison between men and women should be performed.

Data from body composition measured by DEXA and pulmonary function by spirometry should be presented as mean +/- SD in separate tables. Line 194, the authors mentioned table 4 but table 4 presents correlations.

Statistical analysis should be completed with ANCOVA study in each group.

Figure 1, non significant correlation should not be presented.

The authors studied the correlation between parameters. Correlation analyses should be performed in each group (participants without obesity vs participants with obesity). Figure 1, table 4 should be corrected.

Table 1 should be reformatted

Discussion and Conclusion should be revised.

Author Response

Reviewer 1

Comments and Suggestions for Authors

In the manuscript entitled “Association between pulmonary function and body composition in children and adolescents with and without obesity”, the authors presented data from control and obese participants. The study should be strengthened with new and more relevant analyses. Additionalanalyses are needed.

Material and methods

The authors should check the number of participants that varied from 69 (line 55) to 70. In table 1, the authors mentioned 41 participants with obesity and 29 controls whereas in table 2 only 28 controls are studied.

Reply: The authors thank the reviewer for the important feedback. We corrected the number. In brief, we included the following excerpt:

Abstract: “We performed a cross-sectional study with 69 participants, 41 (59.42%) of whom have obesity.

Material and methods: “Initially, the sample consisted of 83 participants. Fourteen participants were excluded, and the final sample comprised 69 participants, 41 (59.42%) of whom had obesity.

Results: We corrected Table 1.

Line 53: The authors mentioned that the study was performed in both sex. In Table 1, data from female should be added.

Reply: We thank the reviewer for the important suggestion. In this context, we added the number of female participants (percentage) in Table 1.

Line 76, z score should be defined and explain why they use it. The authors should explain the interest of using z-score and why it appears to be negative in the control group.

Reply: BMI z-score is used in paediatric population to define nutritional status, according to World Health Organization (WHO) references. The normal BMI z-score range from–2 to +1.

(de Onis M, Onyango AW, Borghi E, Siyam A, Nishida C, Siekmann J. Development of a WHO growth reference for school-aged children and adolescents. Bull World Health Organ. 2007;85(9):660-7. doi: 10.2471/blt.07.043497).

To calculate BMI, we used the calculation: weight / height². Then, according to WHO references, we classify z-score, as stated in the following charts:

The software WHO Anthro Plus, cited in the manuscript, was used to calculate it automatically.

We added information to the manuscript.

In general, comparison between men and women should be performed.

Reply: The authors thank the reviewer for the contribution. In the new version of the manuscript, we included Tables 6 and 7 with the association between body composition, the diagnosis of obesity, and sex. We improved the excerpt of limitations.

Data from body composition measured by DEXA and pulmonary function by spirometry should be presented as mean +/- SD in separate tables. Line 194, the authors mentioned table 4 but table 4 presents correlations.

Reply: We included the information in Tables 5 to 7.

Statistical analysis should be completed with ANCOVA study in each group.

Reply: Dear reviewer, we included several new statistical analyses. However, we did not include ANCOVA. In our study, we did not find any differences between groups for the baseline features. Also, lung function was adjusted by sex and age. In addition, we compared the body composition by groups (obesity status) and grouped by sex. In addition, we can include the ANCOVA, but we need to have support about which markers to include in the model. Thank you so much for correcting our article and improving it.

Figure 1, non significant correlation should not be presented.

Reply: The authors thank reviewer 1 for the contribution. In the new version of the manuscript, we kept the Figure (as recommended by reviewer 2). However, we added the results grouped by obesity status.

The authors studied the correlation between parameters. Correlation analyses should be performed in each group (participants without obesity vs participants with obesity).

Reply: Again, the authors thank Reviewer 1 for the contribution. We believe the manuscript was improved based on his suggestions. We included the correlations analyses in Table 4: -a) overall (presented in the first version of the manuscript); -b) participants with obesity; and -c) participants without obesity.

Figure 1, table 4 should be corrected.

Reply: We corrected Figure 1 and Table 4 as recommended. Please, see the previous questions.

Table 1 should be reformatted

Reply: We reformatted Table 1.

Discussion and Conclusion should be revised.

Reply:  We corrected discussion and conclusion.

Reviewer 2 Report

The authors stated purpose of the study was accomplished. Figure 1 provides interesting and relevant data. Since the data in Figure 2 is stated as no difference in groups with and without obesity, Figure 2 adds nothing to the presentation and can be left out of a revision since the text is clearly stated. The results of Table 2 are adequately summarized in the text and no differences were there, so Table 2 add no further information to the text. Reference to Table 1 should be indicated parenthetically after an initial relevant statement about the content

Pardos can be replaced with an English word, mixed race. 

Author Response

Reviewer 2

Comments and Suggestions for Authors

The authors stated purpose of the study was accomplished.

Figure 1 provides interesting and relevant data.

Reply: The authors thank the reviewer for the important contribution. We performed minor corrections in Figure 1 with the inclusion of the statistical analysis by groups as recommended by reviewer 1.

Since the data in Figure 2 is stated as no difference in groups with and without obesity, Figure 2 adds nothing to the presentation and can be left out of a revision since the text is clearly stated.

Reply: Dear reviewer 2, Figure 2 is important to demonstrate that both groups had equal pubertal development according to genitalia, breasts, and pubic hair.

The results of Table 2 are adequately summarized in the text, and no differences were there, so Table 2 add no further information to the text.

Reply: Dear reviewer, in Table 2 we presented the classification and comparison of the groups regarding the practice of physical activities, according to the IPAQ questionnaire. Curiously, in this case, it is important to present no statistically significant result, because we can prove that both groups had the same profile regarding physical activities, which did not compromise our findings regarding body mass composition and lung function parameters.

Reference to Table 1 should be indicated parenthetically after an initial relevant statement about the content

Pardos can be replaced with an English word, mixed race. 

Reply: The authors thank the reviewer for the important contribution. In this context, we replaced the word “Pardos” with “mixed race”, and we included a minor comment in the Table legend.

Round 2

Reviewer 1 Report

The authors have done a great job to respond to previous comments and took the comments seriously. The manuscript was significantly improved.